# Effects of an Aquaporin 4 Inhibitor, TGN-020, on Murine Diabetic Retina

**DOI:** 10.3390/ijms21072324

**Published:** 2020-03-27

**Authors:** Shou Oosuka, Teruyo Kida, Hidehiro Oku, Taeko Horie, Seita Morishita, Masanori Fukumoto, Takaki Sato, Tsunehiko Ikeda

**Affiliations:** 1Department of Ophthalmology, Osaka Medical College, Takatsuki, Osaka 569-8686, Japan; s_osuka_0606@yahoo.co.jp (S.O.); hidehirooku@aol.com (H.O.); opt168@osaka-med.ac.jp (T.H.); opt094@osaka-med.ac.jp (M.F.); opt147@osaka-med.ac.jp (T.S.); tikeda@osaka-med.ac.jp (T.I.); 2Osaka Kaisei Hospital, Yodogawa Ward, Osaka 532-0003, Japan; infinity_s2000@yahoo.co.jp

**Keywords:** aquaporin 4 (AQP4), TGN-020, reactive oxygen species (ROS), vascular endothelial growth factor (VEGF), Müller cell, diabetic macular edema

## Abstract

Purpose: To investigate the effect of a selective aquaporin 4 (AQP4) inhibitor, 2-(nicotinamide)-1,3,4-thiadiazole (TGN-020), on the expression of vascular endothelial growth factor (VEGF) and reactive oxygen species (ROS) production, as well as on the retinal edema in diabetic retina. Methods: Intravitreal injections of bevacizumab, TGN-020, or phosphate-buffered saline (PBS) were performed on streptozotocin-induced diabetic rats. Retinal sections were immunostained for anti-glial fibrillary acidic protein (GFAP), anti-AQP4, and anti-VEGF. Protein levels of VEGF from collected retinas were determined by Western blot analysis. In addition, retinal vascular leakage of Evans Blue was observed in the flat-mounted retina from the diabetic rats in the presence or absence of TGN-020. Volumetric changes of rat retinal Müller cells (TR-MUL5; transgenic rat Müller cells) and intracellular levels of ROS were determined using flow cytometry analysis of ethidium fluorescence in the presence or absence of TGN-020 or bevacizumab under physiological and high glucose conditions. Results: In the diabetic retina, the immunoreactivity and protein levels of VEGF were suppressed by TGN-020. AQP4 immunoreactivity was higher than in the control retinas and the expressions of AQP4 were co-localized with GFAP. Similarly to VEGF, AQP4 and GFAP were also suppressed by TGN-020. In the Evans Blue assay, TGN-020 decreased leakage in the diabetic retinas. In the cultured Müller cells, the increase in cell volumes and intracellular ROS production under high glucose condition were suppressed by exposure to TGN-020 as much as by exposure to bevacizumab. Conclusion: TGN-020 may have an inhibitory effect on diabetic retinal edema.

## 1. Introduction

Diabetic retinopathy remains a major vision-threatening disease with the following clinical conditions: proliferative retinopathy and macular edema [1,2,3]. As a molecular mechanism for the pathogenesis and development of diabetic retinopathy, hyperglycemia-induced intracellular metabolic disorder including the excessive activity of vascular endothelial growth factor (VEGF) in the retina, oxidative stress such as reactive oxygen species (ROS) produced both intracellularly and extracellularly, accumulation of advanced glycation end-products, and increased cytokines among others are included. Generally, the major cause of visual impairment in patients with diabetic retinopathy is diabetic macular edema (DME) [4]. VEGF is already known to play an important role in the development of DME, and anti-VEGF therapy, which blocks the action of VEGF, is widely used for the treatment of DME [5,6]. Although anti-VEGF therapies are effective, the pathological processes of DME are highly complex, and around 30% of patients with DME are resistant to the treatment [7]. Both VEGF and ROS are essential for physiological function; however, diabetes leads to overproduction of ROS through an upregulation of NADPH oxidase [8,9]. VEGF and ROS also can be mediators of angiogenesis signaling, and excessive ROS causes vascular endothelial dysfunction [10,11]. In addition, oxidative stress is closely associated with diabetic angiogenesis caused by VEGF [12]. Thus, ROS may also be involved in the pathogenesis of VEGF-related DME.

Aquaporin 4 (AQP4) is the most abundant osmotically driven transmembrane water channel, which is essential in normal retinal fluid and water homeostasis [13]. AQP4 is chiefly expressed on the end-feet of astrocytes, which envelope the capillaries in the brain and retina; loss of astrocytes has been reported to cause microvascular damage. Papadopoulos et al. reported that AQP4 not only participates in edema formation, but also in the absorption of excess brain water [14]. Thus, AQP4 is involved in both the vasogenic and cytotoxic edema in the brain. Kaur et al. showed that hypoxic insults cause upregulation of VEGF and AQP4 in brain astrocytes, suggesting that there is a close link between VEGF and AQP4 [15]. In the eye, VEGF and AQP4 are expressed in astrocytes and retinal Müller cells [16,17]. However, in reactive Müller cells in diabetic conditions, there is redistribution of AQP4 expression. This alteration in the AQP4 expression may lead to retinal edema [18]. In addition, expression of AQP4 in astrocytes and Müller cells is increased by nitric oxide (NO). Moreover, NO also causes oxidative stress and death of neurons. Interestingly, another paper by Bi et al. showed that hydrogen peroxide (H_2_O_2_) induces a significant increase in the AQP4 levels in brain astrocytes, and elimination of the oxidative stress depresses the increase [19]. They also showed that H_2_O_2_ increases AQP4 plasma membrane expression and that this change is independent of de novo AQP4 synthesis. These results indicate there is also a close implication between AQP4 and ROS. Therefore, AQP4 has been suggested as a novel therapeutic target for the treatment of diabetic retinopathy, while the redox state is also important [20,21]. AQP4 would also be involved in diabetic retinal edema through vasogenic and cytotoxic mechanisms. Recently, we reported the possibility that AQP4 and VEGF are involved in Müller cell swelling, using diabetic rats and cultured Müller cells [22]. To our knowledge, there is no report related with ROS and AQP4 in diabetic rat retinas.

Taken together, it seems that VEGF, ROS, and AQP4 are closely linked in the retina, and that the unbalance between them may contribute to the development of diabetic retinopathy and DME.

Several chemical structures with an AQP4 inhibitory effect have been identified. Using an in vitro bioassay, 2-(nicotinamide)-1,3,4-thiadiazole (TGN-020) was found to be the most potent AQP4 inhibitor among the agents studied by Huber et al. [23]. In mice with cerebral ischemia, TGN-020 decreased 10% in the brain volume of AQP4-mediated edema [24]. Thus, we hypothesized that AQP4 is involved in diabetic retinal edema and inhibition of the AQP4 channels are promising strategies for DME. To test this hypothesis, we investigated the effects of TGN-020 through the expression of VEGF and AQP4 in diabetic rat retinas in an in vivo study, and then we also performed in an in vitro study to make clear the changes in the volume of Müller cells and their production of ROS under high glucose conditions using flow cytometry. 

## 2. Results

### 2.1. Immunohistochemistry of Retinal Slices

Photomicrographs of retinal sections stained immunohistochemically with an anti-VEGF antibody (green) and an anti-glial fibrillary acidic protein (GFAP) antibody (red) are shown in Figure 1. Sections were obtained from eyes receiving intravitreal injections of (a) a vehicle in control rats, and (b) a vehicle or (c) bevacizumab or (d) TGN-020 in streptozotocin (STZ)-induced diabetic rats. Compared with the vehicle-injected control rats (Figure 1a), immunoreactivity to VEGF was increased in the STZ-induced diabetic rats (Figure 1b). Immunoreactivity to VEGF was increased in the ganglion cell layer (GCL), the inner plexiform layer (IPL), the inner nuclear layer (INL), and the outer plexiform layer (OPL). The VEGF expression was intensified in these internal retinal layers. Immunoreactivity to GFAP, a glial cell marker, was also increased in the diabetic rats. Double staining for VEGF and GFAP demonstrated that the expression of VEGF was co-localized with GFAP expression. These reactions were reduced in not only bevacizumab-injected (Figure 1c) but also TGN-020-injected (Figure 1d) STZ-induced diabetic rats. Photomicrographs of retinal sections stained immunohistochemically with an anti-AQP4 antibody (red) and an anti-GFAP antibody (green) are shown in Figure 2. AQP4 and GFAP were enhanced in STZ-induced diabetic rats compared to control rats (Figure 2a,b), and this enhancement was reduced not only in the bevacizumab-injected group (Figure 2c) but also in the TGN-020-injected diabetic rats (Figure 2d). Additionally, the observed longitudinal pattern of immunoreactivity to AQP4 in the IPL, INL, OPL, and the outer nuclear layer (ONL) suggested that AQP4 expression was increased in the Müller cells.

### 2.2. Protein Levels of VEGF in Retinas by Western Blot

Figure 3 shows retinal VEGF protein levels determined by Western blot analysis. The levels of VEGF were normalized to the level of α-tubulin. VEGF expression in STZ-induced diabetic rats significantly increased by 260.5% compared to in non-diabetic rats. Furthermore, among diabetic rats, the protein levels of VEGF decreased by 35% in the TGN-020-injected group and by 54.2% in the bevacizumab-injected group compared to the vehicle-injected group (*p* < 0.0001, Scheffe).

### 2.3. Measurement of Retinal Thickness

Table 1 shows the retinal thicknesses in the age-matched control rats, the vehicle-injected diabetic rats, and the TGN-020-injected diabetic rats. The average of total retinal thickness and each layer’s thickness were calculated and are shown. All values are expressed as the mean ± standard deviation. The mean total retinal thickness was significantly thicker in the vehicle-injected diabetic rats than in the control rats. In the changes of individual retinal layer thickness, there was a significant increase in all layers except for OPL and the layer between the retinal internal limiting membrane (ILM) and INL. The increase in retinal thickness observed in the diabetic rats was suppressed in the TGN-020-injected diabetic rats. 

### 2.4. Images of Evans Blue Injected Flat-Mounted Retinas by Epifluorescence Microscope

Retinal blood vessel permeability in the STZ-induced diabetic rats in the presence or absence of TGN-020 is shown in Figure 4. In both groups, the leakage was observed at the vascular bifurcation, but the extravasations seemed to be suppressed by administration of TGN-020. 

### 2.5. Cellular Volume and Intracellular Levels of ROS by Flow Cytometry

Figure 5 shows graphs illustrating the changes in the cellular volumes of rat retinal Müller cells (TR-MUL5) that were cultured in high glucose (25 mM) medium or physiological concentration of glucose (5.5 mM) medium. On the basis of this flow cytometry, the volumes of the cells cultured in the high glucose medium were increased significantly compared to the volumes of those cultured in the physiological glucose medium. On the other hand, in the culture with the high glucose medium, the addition of TGN-020 suppressed the increase in cell volume, similar to the addition of bevacizumab (*p* < 0.05).

Figure 6 shows graphs illustrating changes in intracellular levels of ROS in TR-MUL5 under the high glucose (25 mM) or physiological concentration of glucose (5.5 mM) medium. We examined the generation of ROS in TR-MUL5 cells by adding hydroethidine and measuring ethidium fluorescence by flow cytometry. The ethidium fluorescence in the high glucose medium was significantly higher than in the physiological glucose medium. In the culture with the high glucose medium, the addition of TGN-020 suppressed the increase in ROS production as well as the addition of bevacizumab. 

### 2.6. Immnostaining of Cultured Müller Cells (TR-MUL5) 

Representative photomicrographs of immunostaining with anti-VEGF (green) and anti-AQP4 antibodies (red) in the TR-MUL5 cells are shown in Figure 7. Immunoreactivities to VEGF and AQP4 were enhanced in cells cultured in high glucose medium compared to those of low glucose medium. The increased reactivity in high glucose condition was depressed by TGN-020.

## 3. Discussion

This study demonstrated that the intravitreal injection of TGN-020 could suppress retinal edema in diabetic rat retinas. The in vivo study using diabetic rats showed that the immunoreactivity and protein levels of VEGF were increased and that TGN-020 depressed the increase. In addition, immunoreactivities to VEGF, AQP4, and GFAP were also increased in diabetic rat retinas, and these expressions were well co-localized. In vitro study showed high glucose condition caused an increase in volumes of Müller cells and intracellular ROS production and that TGN-020 depressed these increases in Müller cells as much as bevacizumab. 

Hyperglycemia affects retinal vessels and neuroglia. Not only angiopathy but even neurodegeneration begins at the early stage of diabetic retinopathy [10]. Our in vivo study from rat retinal slices showed increased retinal thickness along with increased GFAP expressions that transverse longitudinally throughout the retinal layers, suggesting that the Müller cells are reactivated. Increased expressions of VEGF and AQP4 with GFAP suggested that reactive Müller cells express these proteins in diabetic retinas and that they occur during a course of neurodegeneration. Thus, Müller cells may play a crucial role in the development of diabetic retinal edema, as has been shown in ischemic retinal edema [25,26]. Additionally, the rat retinal slices showed significantly thicker mean total retinal thickness in diabetic rats, and the increased retinal thickness was decreased by intravitreal injection of TGN-020. Results from our in vivo study suggested that hyperglycemia-induced increase in the retinal thickness was related to reactive Müller cells, as well as the fact that AQP4 channels were responsible for the edema formation. From a histological point of view, confirmation of our findings that intravitreal injection of TGN-020 can suppress the retinal thickness in diabetic rats would be informative.

On the other hand, the in vitro study showed that the volume of Müller cells under high glucose media significantly increased compared to the control, as well as the fact that this increase of the cell volume was suppressed by TGN-020. Under normal circumstances, Müller cells move the excess water in the retina into the blood by transcellular water transport through AQP4 channels [13]. However, in a diabetic condition, Müller cell dysfunction causes dysregulation of water movement across the gliovascular interface, resulting in osmotic expansion of Müller cells. The Müller cell swelling in this mechanism is known as cytotoxic edema, and this kind of glial swelling is recognized as a main cause of ischemic brain edema [27]. The swelling of Müller cells obtained from our flow cytometry analyses in our present study indicates that diabetic induced retinal edema may involve cytotoxic edema. 

Besides the cytotoxic edema characterized by swelling of Müller cells, vasogenic edema should be considered in the development of retinal edema [28]. In this regard, we observed the effects of TGN-020 against the hyperglycemia-induced retinal vascular leakage by using the Evans Blue injected flat-mounted retinas of diabetic rats. The leakage seemed to be decreased by TGN-020. Unfortunately, we could not quantify the degree of the leakage to compare the degree of vascular leakage in the presence or absence of TGN-020; however, this retinal vascular leakage in the diabetic rats was possibly thought to be associated with hyperglycemia-induced vasogenic edema. 

To determine the mechanisms by which TGN-020 suppresses diabetic retinal edema, we explored in vitro study using a TR-MUL5 line, a transformed cell line of Müller cells from rats. Our in vitro study mimicked our in vivo study in that high glucose condition increased the cellular volume and TGN-020 depressed the increase. We previously showed that TGN-020 depressed the VEGF-induced cellular swelling of TR-MUL5 cells, indicating that VEGF-induced increase of cellular volume is mediated by AQP4 [22]. Consistently, high glucose condition caused swelling of Müller cells, and ROS was increased in these cells. Because TGN-020 depressed these changes under high glucose condition, ROS might be mediated Müller cell swelling through AQP4 channels. ROS and AQP4 are closely linked. It has been reported that excessive oxidative stress and mitochondrial permeability transition leads to ammonia-induced astrocyte swelling in the brain by using an in vivo model of hepatic encephalopathy [29]. In another study, endothelin-2 was found to inhibit NADPH oxidase, and endothelin-2-injected mice with injured blood-retinal barrier observed in diabetic retinopathy showed increased expressions of AQP4 in the retina [30]. To our knowledge, there are only a few papers on the topic of the increase of reactive AQP4 on ROS production, and our present study provides a new finding using a flow cytometry study in rat retinal Müller cells. 

There are some limitations in this study. First, we used rat retinas, not mammal retinas. In diabetic rats, an increase in retinal thickness was observed in all layers except the layer between ILM and INL, but Bandello et al. reported an increase in retinal thickness in INL and OPL in human diabetic retinal edema [3]. Thus, it is unclear whether humans can recognize the similarly effect of reducing the retinal thickness as obtained this time. The reason for this difference is that the rat retina was used in this study, but the rat retina does not have a macular region, and the histological characteristics of the retina may be different. Toyoda et al. also reported that thickening of the entire retina was observed in Spontaneously Diabetic Torii (SDT) rats of spontaneous diabetes model, and thus water may easily escape into the interstitial spaces of the rat retina [31]. It will be important to confirm the present results using mammals that possess a macula, such as monkeys. Second, we conducted the in vitro study using a rat retinal Müller cell line, not primary Müller cells. To confirm these data, we need to perform similar experiments using primary Müller cells isolated from the rats. Third, in the in vivo study, the retinal vascular leakage of the Evans Blue was observed in the flat-mounted retinas from diabetic rats. We detected the leakage in diabetic rats and the leakage seemed to be decreased by TGN-020; however, we could not determine the quantity of the leakage in our present study. Fourth, we could not evaluate AQP4 levels assessed by Western blot and ROS levels detected in tissue from animals using the probe 2,7-dichlorofluoroscein (DCF). The above study may be necessary to clarify the relationship between VEGF, AQP4, and ROS for DME. These studies would be necessary to clarify the relationship between VEGF, AQP4, and ROS for DME. 

In conclusion, our findings from in vivo and in vitro studies showed that the intravitreal injection of TGN-020 can be a potential therapy to suppress diabetic retinal edema. Interestingly, overproduction of ROS in swelling Müller cells induced by high glucose was suppressed by TGN-020, an AQP4 inhibitor. TGN-020 may have an inhibitory effect on diabetic retinal edema from both aspects of glial degeneration and vessel-barrier dysfunction that occur in diabetic retinopathy.

## 4. Materials and Methods 

### 4.1. Animals

Nine-week-old male Wistar rats were obtained from Japan SLC Inc. (Shizuoka, Japan). All rats were housed at room temperature and had free access to food and water. All experiments were performed in accordance with the ARVO Statement for the Use of Animals in Ophthalmic and Vision Research. The experimental procedures were approved by the Osaka Medical College Committee on the Use and Care of Animals (Approval number: 2019-100). 

### 4.2. Chemicals

All chemicals were obtained from Sigma-Aldrich (St. Louis, MO, USA), unless otherwise specified.

### 4.3. Induction of Diabetes

To induce diabetes, each animal received a single 40 mg/kg intravenous injection of streptozotocin in a 10 mM sodium citrate buffer, pH 4.5, after an overnight fast while under general anesthesia. As controls, nondiabetic animals received an injection of citrate buffer alone. Animals with blood glucose levels higher than 250 mg/dL 24 h after injections were considered diabetic. All experiments were conducted 8 weeks after the induction of diabetes.

### 4.4. Intravitreal Injection

Two microliter intravitreal injections of TGN-020 (0.02 mg/μL) or bevacizumab (0.025 mg/μL) (Genentech Inc, San Francisco, CA, USA) or vehicle alone (phosphate-buffered saline (PBS), pH 7.4) were performed on STZ-induced diabetic rats using a Hamilton syringe and a 30-gauge needle. Animals received general anesthesia, and perfused fixation was performed 48 h after the intravitreal injections. 

### 4.5. Immunohistochemistry of Retinal Slices

Eight rats were deeply anesthetized by an intraperitoneal injection of pentobarbital sodium (25 mg/kg body weight, Nembutal, Kyoritsu Pharmaceutical Co., Ltd., Tokyo, Japan), and perfused through the heart with saline followed by 4% paraformaldehyde (PFA) in a 0.1 M PBS, pH 7.4. The retinas were carefully removed and post-fixed in 4% PFA in PBS overnight. After washing with PBS, the retinal tissues were immersed in 30% sucrose for 2–3 days at 4 °C and then embedded in OCT (optimal cutting temparature) compound (Sakura Finetechnical, Tokyo, Japan). Then, 4 μm frozen sections were cut with a cryostat (CM3050S, Leica, Wetzlar, Germany). After blocking with 1% normal goat serum plus 1% BSA (bovine serum albumin) and 0.1% Triton X-100 in PBS, the retinal sections were incubated overnight at 4 °C with the following primary antibodies: mouse monoclonal anti-VEGF (1:500, Santa Cruz Biotechnology, Inc., Dallas, TX, USA), rabbit polyclonal anti-GFAP (1:500, Merck Millipore, Billerica, MA, USA), and anti-AQP4 (1:500, Santa Cruz). These sections were incubated for 2 h at 37 °C in Alexa 594 or Alexa 488-conjugated to the appropriate secondary antibodies (1:1000; Invitrogen, Carlsbad, CA, USA). The processed sections were photographed with a fluorescence microscope (BZ-X700, Keyence, Osaka, Japan). 

### 4.6. Protein Levels of VEGF in Retinas by Western Blot

Retinas were excised from the eyes and homogenized in a lysis buffer containing 1 mM phenyl methanesulfonyl fluoride, 10 μM pepstatin A, 10 μM leupeptin, 10 μM aprotinin, 0.1% sodium dodecyl sulfate (SDS), 1% Nonidet P-40, 5% sodium deoxy cholate, 50 mM Tris-HCl (pH 7.6), and 150 mM sodium chloride. The suspension was centrifuged, and the total protein concentration of the resulting supernatant was determined using the Lowry method (DC Protein Assay Reagent, Bio-Rad, Hercules, CA, USA). Samples were separated on a 12% SDS-polyacrylamide gel and blotted onto PVDF (polyvinylidene difluoride) membranes. The membranes were then blocked with 5% skim milk in Tris-buffered saline, pH 7.4, with 0.1% Tween 20 (TBS-T), followed by an overnight incubation at 4 °C with a rabbit polyclonal anti-VEGF (147) antibody (Santa Cruz). Tubulin (α-tubulin, 1:1000; Merck Millipore, CP06) was used as an internal control. The membranes were washed in TBS-T followed by incubation with a peroxidase-conjugated goat anti-rabbit IgG (Immunoglobulin G) (1:2500, Promega, Madison, WI, USA) secondary antibody for 2 h at 37 °C. The protein bands were visualized following the addition of an ECL Plus Western blotting detection system (GE Healthcare, Little Chalfont, United Kingdom). Protein band densities were measured with a luminescent image analyzer (LAS-3000, Fujifilm, Tokyo, Japan). Relative protein levels were quantified using the embedded software (Multi Gauge version 3.0) and standardized according to tubulin protein levels.

### 4.7. Measurement of Retinal Thicknesses

Using the frozen retinal sections created in the same way as mentioned above, the retinal thickness was measured by a fluorescent microscope (BZ-X700, Keyence, Osaka, Japan). As previously reported [31], the retinal thicknesses were measured 500, 1000, and 1500 μm from the optic nerve disc using Image J software (National Institutes of Health, Bethesda, MD, USA) and the average of the three values was taken. The total retinal thickness was defined as the distance between the retinal internal limiting membrane (ILM) and the retinal pigment epithelium (RPE). In the retina, the thicknesses between the ILM and the inner nuclear layer (INL), the INL thickness, the OPL thickness, the ONL thickness, and the photoreceptor layer thickness were calculated. All values were expressed as the mean ± standard deviation. The Scheffe’s multiple comparison procedure was used for comparisons between three groups. *p* < 0.05 was considered statistically significant.

### 4.8. Evans Blue Assay

Retinal blood vessel permeability of the STZ-induced diabetic rats in the presence or absence of TGN-020 was tested by Evans Blue assay. A total of 100 μL of Evans Blue (20 mg/mL) in PBS was injected into the tail vein. After 10 min, retinas were carefully explanted and post-fixed in 4% PFA in PBS overnight. Flat-mounted retinas were created and extravasation was evaluated by epifluorescence analysis at an excitation wavelength of 594 nm with a laser scanning confocal microscope (TCS SP8, Leica, Tokyo, Japan). 

### 4.9. Cell Culture

A rat retinal Müller cell line, TR-MUL5, was obtained from Fact, Inc. (Sendai, Japan) [32]. These cells were collected from transgenic rats carrying a temperature-sensitive SV40 large T antigen gene [32,33]. The TR-MUL5 cells were cultured in Dulbecco’s modified Eagle’s medium (DMEM) supplemented with 10% fetal bovine serum (FBS) at 33 °C in a humidified atmosphere of 5% CO_2_/air. Before reaching confluence, the media was changed to the control media (5.5 mM glucose in DMEM) or the high glucose media (25 mM glucose in DMEM), and the temperature was raised to 37 °C in order to arrest the proliferation by reducing the expression of the large T antigen. Additionally, the media was changed to a control DMEM media containing a physiological concentration of 5.5 mM glucose. The TR-MUL5 cells were then cultured in control mediums lacking 10% FBS and serum-deprived overnight. TR-MUL5 cells at passages 18 and 20 were used in this study. Detailed procedures and treatment protocols for each experiment are described later.

### 4.10. Measurement of Cell Volume and Intracellular Levels of ROS Production by Flow Cytometry

To examine the changes in the volume of TR-MUL5 cells and intracellular production of ROS under high glucose condition, the cells were incubated in high glucose (25 mM) or physiological concentration of low glucose (5.5 mM) medium for 2–3 days. Then, volumetric changes of TR-MUL5 and intracellular levels of ROS were determined using flow cytometry analysis of ethidium fluorescence in the presence or absence of TGN-020 (100 nM) or bevacizumab overnight under physiological and high glucose conditions. The level of superoxide in TR-MUL5 cells was measured using hydroethidine, a fluorogenic probe. Hydroethidine is oxidized by superoxide to form ethidium, a fluorescent product, which is then retained intracellularly allowing a semiquantitative estimation of intracellular superoxide levels [34,35]. Cells were harvested via trypsinization and centrifuged at 800 *g* for 5 min. After washing with PBS, the cells were resuspended in phenol red-free DMEM with hydroethidine (1 μg/mL) for 30 min at 37 °C. Cell densities were adjusted to 2.0 × 10^5^ cells/mL. The changes in cellular volume of Müller cells and the intracellular levels of superoxide were analyzed using flow cytometry (EC800 Analyzer, Sony Biotechnology, Inc., Tokyo, Japan) with 488 nm excitation and 590 to 610 nm emission wavelengths [35]. The acquisition and analysis software on the EC800 was used to acquire and quantify the fluorescent intensities. We previously described the details of determining the volume changes of TR-MUL5 cells using an EC800 flow cytometry analyzer [36,37]. 

### 4.11. Immunostaining of Cultured Müller Cells

To determine the effect of TGN-020 on the expressions of VEGF and AQP4 in the TR-MUL5 cells, the cells that were incubated in high glucose (25 mM) media with and without TGN-020 or physiological concentration of low glucose (5.5 mM) media were examined by immunocytochemistry. After fixation by 4% formaldehyde, the cells were incubated with primary antibodies of rabbit polyclonal anti-AQP4 and mouse monoclonal anti-VEGF (1:100; Santa Cruz Biotechnology, Inc., Dallas, TX, USA) overnight at 4 °C. After rinsing by PBS and blocking, these cells were incubated for 2 h at room temperature in Alexa 594 or Alexa 488-conjugated to the appropriate secondary antibodies (Invitrogen, Carlsbad, CA, USA) diluted by 1:500. Nuclei were stained with 4′,6-diamidino-2-phenylindole (DAPI, 1:1000, Dojindo, Inc., Kumamoto, Japan.). The processed samples were photographed with a fluorescence microscope (BZ-X700, Keyence, Osaka, Japan).

## Figures and Tables

**Figure 1 ijms-21-02324-f001:**
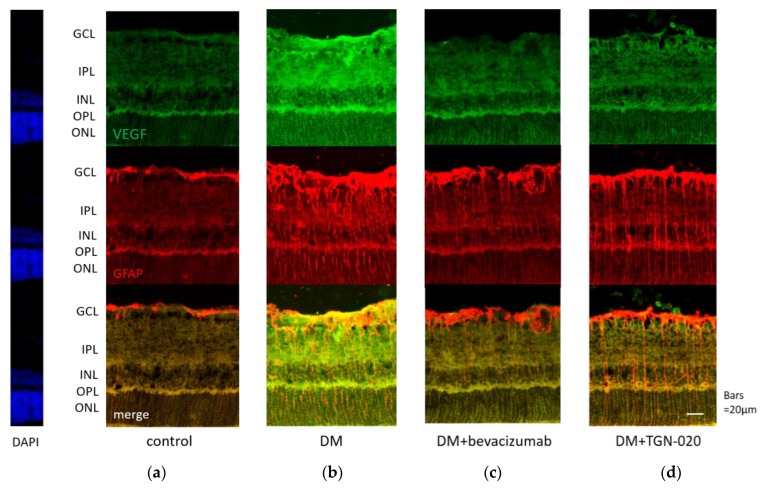
Representative photomicrographs of retinal tissues stained immunohistochemically with anti-vascular endothelial growth factor (VEGF) (**green**) and anti-glial fibrillary acidic protein (GFAP) (**red**) antibodies. Sections were obtained from eyes receiving intravitreal injections of (**a**) a vehicle in control rats, and (**b**) a vehicle or (**c**) bevacizumab or (**d**) 2-(nicotinamide)-1,3,4-thiadiazole (TGN-020) in streptozotocin (STZ)-induced diabetic rats. Compared with the vehicle-injected control rats (**a**), immunoreactivity to VEGF was increased in the vehicle-injected STZ-induced diabetic rats (**b**). This change was suppressed in the TGN-020 injected group (**d**), as well as in the bevacizumab-injected group (**c**). The expression of VEGF was co-localized with GFAP expression.

**Figure 2 ijms-21-02324-f002:**
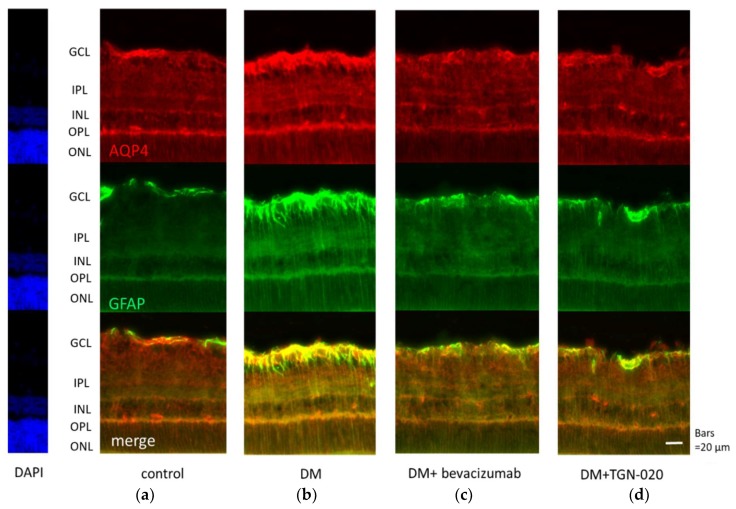
Representative photomicrographs of retinal tissues stained immunohistochemically with anti-aquaporin 4 (AQP4) (**red**) and anti-GFAP (**green**) antibodies. Sections were obtained from eyes that had received intravitreal injections of (**a**) a vehicle in control rats, and (**b**) a vehicle or (**c**) bevacizumab or (**d**) TGN-020 in STZ-induced diabetic rats. AQP4 and GFAP were enhanced in STZ-induced diabetic rats (**b**) compared to control rats (**a**), and this change was reduced not only in the bevacizumab-injected group (**c**) but also in the TGN-020-injected group (**d**).

**Figure 3 ijms-21-02324-f003:**
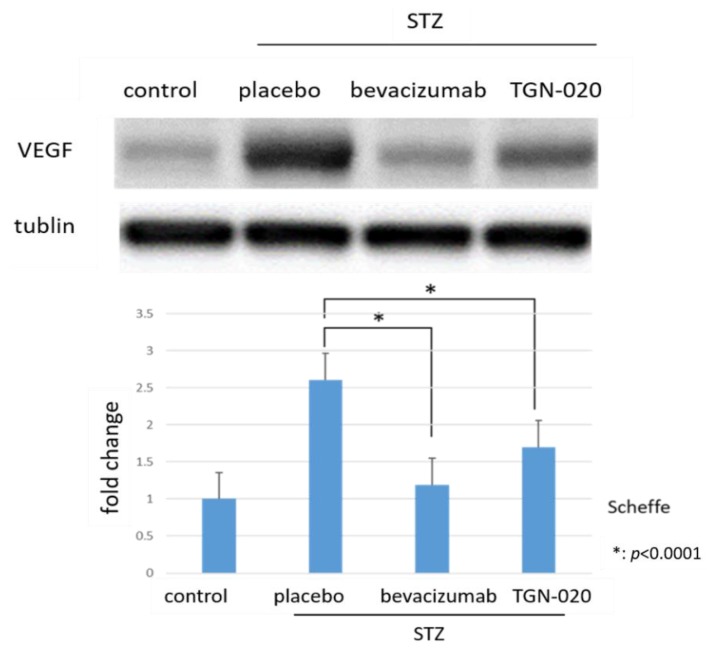
Protein levels of VEGF in retinas by Western blot analyses. Compared to non-diabetic rats, VEGF expression in STZ-induced diabetic rats significantly increased. Additionally, this increase of VEGF decreased in the TGN-020-injected diabetic rats as well as in the bevacizumab-injected group (*p* < 0.0001).

**Figure 4 ijms-21-02324-f004:**
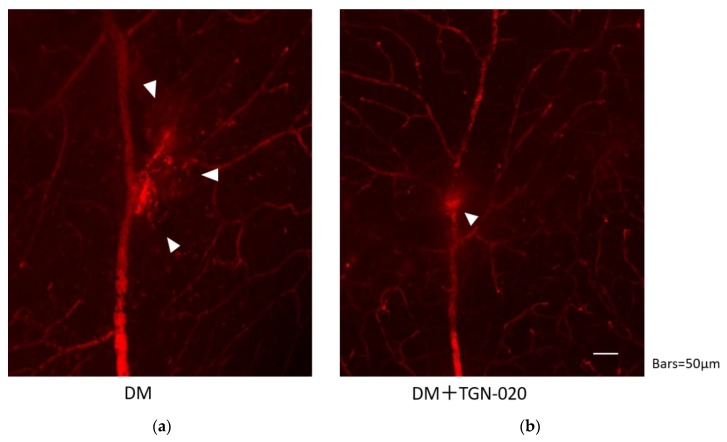
Retinal blood vessel permeability in the STZ-induced diabetic rats in the absence (**a**) or presence (**b**) of TGN-020. Extravasations are highlighted by arrowheads. In both groups, extravasations were observed at the vascular bifurcation, but extravasations seemed to be suppressed in the TGN-020-injected group.

**Figure 5 ijms-21-02324-f005:**
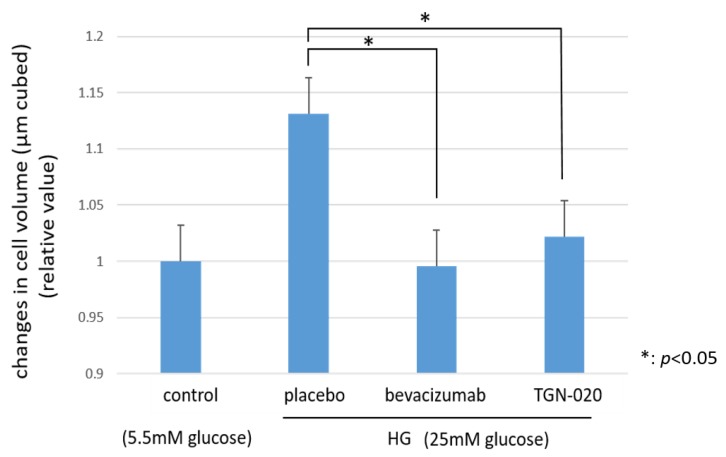
The changes in cellular volumes of rat retinal Müller cells (TR-MUL5 ) that were cultured in high glucose (25mM) medium or physiological concentration of glucose (5.5mM) medium. The cell volumes cultured in the high glucose medium were significantly larger than in the physiological glucose medium. In the culture with the high glucose medium, the addition of TGN-020 suppressed the increase in cell volume as well as the addition of bevacizumab.

**Figure 6 ijms-21-02324-f006:**
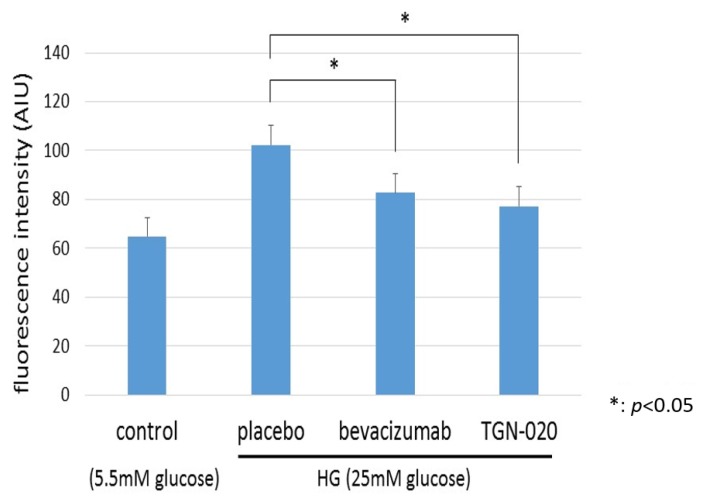
The changes in reactive oxygen species (ROS) production of TR-MUL5 cells that were cultured in high glucose (25 mM) or physiological concentration of glucose (5.5 mM) condition. The fluorescence intensity in the high glucose medium was significantly higher than in the physiological glucose medium. In the culture with high glucose medium, the addition of TGN-020 suppressed the increase in cell volume as well as the addition of bevacizumab.

**Figure 7 ijms-21-02324-f007:**
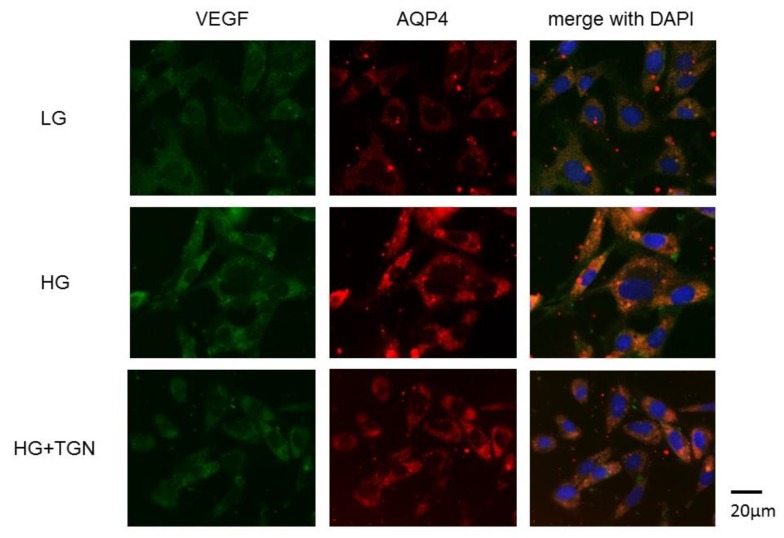
Immnostaining of VEGF and AQP4 in TR-MUL5 cells. Expressions of VEGF and AQP4 were intensified in cells cultured in high glucose medium (HG) compared to those cultured in low glucose medium (LG). This enhancement seemed to be reduced by TGN-020 (HG + TGN).

**Table 1 ijms-21-02324-t001:** The retinal thicknesses in the age-matched control rats, the vehicle-injected diabetic rats, and the TGN-020-injected diabetic rats. The mean total retinal thickness was significantly thicker in the diabetic rats than in the control rats. In the changes of individual retinal layer thickness, there was a significant increase in all layers except for outer plexiform layer (OPL) and the layer between internal limiting membrane (ILM) and inner nuclear layer (INL).

Thickness (μm)	Control	DM	DM + TGN-020
Total retina	208.1 ± 19.6	253.7 ± 26.2 **	209.1 ± 33.1
Between ILM and INL	75. 7 ± 13.5	81.2 ± 9.8	69.0 ± 17.8
INL	31.3 ± 3.1	40.1 ± 5.0 **	33.2 ± 4.4
OPL	16.3 ± 6.7	18.1 ± 9.6	10.2 ± 7.1
ONL	57.7 ± 4.9	70.4 ± 6.8 **	58.0 ± 5.3
Photoreceptor layer	37.3 ± 5.0	44.3 ± 5.4 **	38.7 ± 6.4

** *p* < 0.01 (Scheffe) (mean ± standard deviation (SD)). DM; diabetic rat retina, TGN—020; 2-(nicotinamide)-1,3,4-thiadiazole, an AQP4 inhibitor, ILM; internal limiting membrane, INL; inner nuclear layer, OPL; outer plexiform layer, ONL; outer nuclear layer.

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
