# Peer review of "Effects of an Aquaporin 4 Inhibitor, TGN-020, on Murine Diabetic Retina"

_ijms, 2020, doi:10.3390/ijms21072324_

Round 1

Reviewer 1 Report

The manuscript “Effects of an Aquaporin 4 Inhibitor, TGN-020, on Diabetic Retina” by Shou Oosuka and collegues investigated the effects of a selective aquaporin 4 inhibitor, TGN-020, on the expression of VEGF and ROS production, as well as on the retinal edema in diabetic retina. This work is the continuation of a paper, previously published by the authors studying an in vitro model, with very similar goals (Graefes Arch Clin Exp Ophthalmol. 2017 Jun;255(6):1149-1157). The manuscript is clear and the results are interesting, but some points should need corrections to improve the overall quality of the manuscript.

  • the title should be changed in "Effects of an Aquaporin 4 Inhibitor, TGN-020, on murine diabetic retina"
  • AQP4 levels should be assessed by western blot in non-diabetic and STZ-induced diabetic rats, similarly to the VEGF expression measurement.
  • ROS levels should be detected in tissue from animals using the probe 2,7-dichlorofluoroscein (DCF) (Invest Ophthalmol Vis Sci. 2019 May; 60(6): 2369–2379).
  • VEGF and AQP4 levels should be also assessed in the TR-MUL5 cells.
  • The manuscript should be improved especially as regards the description and motivation of the experiments on both animals and cells, well describing the rationale for choosing to use both models.

Author Response

The manuscript “Effects of an Aquaporin 4 Inhibitor, TGN-020, on Diabetic Retina” by Shou Oosuka and collegues investigated the effects of a selective aquaporin 4 inhibitor, TGN-020, on the expression of VEGF and ROS production, as well as on the retinal edema in diabetic retina. This work is the continuation of a paper, previously published by the authors studying an in vitro model, with very similar goals (Graefes Arch Clin Exp Ophthalmol. 2017 Jun;255(6):1149-1157). The manuscript is clear and the results are interesting, but some points should need corrections to improve the overall quality of the manuscript.

the title should be changed in "Effects of an Aquaporin 4 Inhibitor, TGN-020, on murine diabetic retina"
AQP4 levels should be assessed by western blot in non-diabetic and STZ-induced diabetic rats, similarly to the VEGF expression measurement.
ROS levels should be detected in tissue from animals using the probe 2,7-dichlorofluoroscein (DCF) (Invest Ophthalmol Vis Sci. 2019 May; 60(6): 2369–2379).
VEGF and AQP4 levels should be also assessed in the TR-MUL5 cells.

The manuscript should be improved especially as regards the description and motivation of the experiments on both animals and cells, well describing the rationale for choosing to use both models.

Response: We greatly appreciate the Reviewer's instructive comments. Please note that we completely agree with the Reviewer's suggestion regarding the title. Please note that we have now changed the title to read as follows:

“Effects of an Aquaporin 4 Inhibitor, TGN-020, on Murine Diabetic Retina”. (Page 1, Lines 2 and 3)

Moreover, we also agree with the Reviewer's suggestion that additional information on AQP4 levels assessed by western blot and ROS levels detected in tissue from animals using the probe 2,7-dichlorofluoroscein (DCF) would be valuable. However, and very regrettably, we are simply unable to perform the experiments, as there are no remaining samples and it takes three months to create an experimental diabetic rat model. We sincerely hope that the Reviewer will accept our sincere apology on this matter. Please note that we have now added the following statements in the Discussion section regarding the limitations of our study:

"Fourth, we did not evaluate AQP4 levels assessed by western blot and ROS levels detected in tissue from animals using the probe 2,7-dichlorofluoroscein (DCF). In addition, we did not evaluate VEGF and AQP4 levels assessed in the TR-MUL5 cells. These studies would be necessary to clarify the relationship between VEGF, AQP4, and ROS for DME." (Page 9, Lines 252-255).

In addition, please note that we enlisted the services of a professional, native-English-speaking medical editor to review and revise the English in our manuscript, and that the Track Changes feature of the Microsoft Word program was used to illustrate the changes.

Reviewer 2 Report

This study investigated the effect of TGN-020, a selective aquaporin 4 (AQP4) inhibitor, on diabetic retina by using streptozotocin (STZ) induced diabetic rat and high glucose cultured Müller cell line (TR-MUL5). The authors found that TGN-020 decreased VEGF expression and vessel leakage in diabetic retinas. The authors also found that ROS production was suppressed by TGN-020 in high glucose cultured Müller cells. The authors conclude that TGN-020 may have an inhibitory effect on diabetic retinal edema.

Page 1, Line 23-25. “AQP4 immunoreactivity was higher than in the control retinas and the expressions of AQP4 were colocalized with GFAP.” This sentence should describe the effect of TGN-020 on AQP4 and GFAP expression.

Author Response

This study investigated the effect of TGN-020, a selective aquaporin 4 (AQP4) inhibitor, on diabetic retina by using streptozotocin (STZ) induced diabetic rat and high glucose cultured Müller cell line (TR-MUL5). The authors found that TGN-020 decreased VEGF expression and vessel leakage in diabetic retinas. The authors also found that ROS production was suppressed by TGN-020 in high glucose cultured Müller cells. The authors conclude that TGN-020 may have an inhibitory effect on diabetic retinal edema.

Page 1, Line 23-25. “AQP4 immunoreactivity was higher than in the control retinas and the expressions of AQP4 were colocalized with GFAP.” This sentence should describe the effect of TGN-020 on AQP4 and GFAP expression.

Response: We greatly appreciate the Reviewer's instructive and helpful comment. Please note that we completely agree with the Reviewer's suggestion. Please note that we have now revised the associated sentence to read as follows:

"Similarly to VEGF, AQP4 and GFAP were also suppressed by TGN-020." (Page 1, Lines 25).

Round 2

Reviewer 1 Report

However, understanding the difficulty of re-creating the in vivo model, it would be appropriate to assess VEGF and AQP4, as previously requested, in TR-MUL5 cells..

Author Response

Reviewer 1
However, understanding the difficulty of re-creating the in vivo model, it would be appropriate to assess VEGF and AQP4, as previously requested, in TR-MUL5 cells.

Response: We greatly appreciate the Reviewer's instructive comment. Therefore, we performed the additional experiments of immnostaining of TR-MUL5 cells to assess the effects of TGN-020 on VEGF and AQP4. We have now added Figure 7 and the following statements in both the Results and the Methods sections:

"2.6 Immnostaining of cultured Müller cells (TR-MUL5)

Representative photomicrographs of immunostaining with anti-VEGF (green) and anti-AQP4 antibodies (red) in the TR-MUL5 cells are shown in Fig. 7. Immunoreactivities to VEGF and AQP4 were enhanced in cells cultured in high glucose medium compared to those of low glucose medium. The increased reactivity in high glucose condition was depressed by TGN-020." (Page 8-9, Lines 184-188).

Figure 7. Immnostaining of VEGF and AQP4 in TR-MUL5 cells. Expressions of VEGF and AQP4 were intensified in cells cultured in high glucose medium (HG) compared to those cultured in low glucose medium (LG). This enhancement seemed to be reduced by TGN-020 (HG+TGN)." (Page 9, Lines 190-192).

"4.11 Immunostaining of cultured Müller cells

To determine the effect of TGN-020 on the expressions of VEGF and AQP4 in the TR-MUL5 cells, the cells which were incubated in high glucose (25 mM) media with and without TGN-020 or physiological concentration of low glucose (5.5 mM) media were examined by immunocytochemistry. After fixation by 4 % formaldehyde, the cells were incubated with primary antibodies of rabbit polyclonal anti-AQP4 and mouse monoclonal anti-VEGF (1:100; Santa Cruz Biotechnology, Inc., Dallas, TX, U.S.A.) overnight at 4° C. After rinsing by PBS and blocking, these cells were incubated for 2 hrs at room temperature in Alexa 594 or Alexa 488-conjugated to the appropriate secondary antibodies (Invitrogen, Carlsbad, CA, USA) diluted by 1:500. Nuclei were stained with 4′,6-diamidino-2-phenylindole (DAPI, 1:1000, Dojindo, Inc., Kumamoto, Japan.). The processed samples were photographed with a fluorescent microscope (BZ-X700, Keyence, Osaka, Japan)." (Page 13, Lines 389-400).

Round 3

Reviewer 1 Report

I thank the authors for the additional cellular experiments.